**Subject Category:**
Biology (whole organism)

behaviour/cognition/ecology

laterality, hemispheric specialization, foraging, crater feeding, bottlenose dolphin, *Tursiops truncatus*

**Author for correspondence:**
J. Daisy Kaplan
e-mail: jdaisykaplan@gmail.com

# Behavioural laterality in foraging bottlenose dolphins (*Tursiops truncatus*)

J. Daisy Kaplan[1], Samantha Y. Goodrich[2],
Kelly Melillo-Sweeting[1] and Diana Reiss[3]

[1]Dolphin Communication Project, Port St Lucie, FL 34985, USA
[2]Department of Psychology, St Mary's College of Maryland, St Mary's City, MD, USA
[3]Department of Psychology, Hunter College, CUNY, New York, NY, 10065, USA

JDK, 0000-0002-9170-8922; KM-S, 0000-0002-1767-1009;
DR, 0000-0002-2308-0483

Lateralized behaviour is found in humans and a wide variety of other species. At a population level, lateralization of behaviour suggests hemispheric specialization may underlie this behaviour. As in other cetaceans, dolphins exhibit a strong right-side bias in foraging behaviour. Common bottlenose dolphins in The Bahamas use a foraging technique termed 'crater feeding', in which they swim slowly along the ocean floor, scanning the substrate using echolocation, and then bury their rostrums into the sand to obtain prey. The bottlenose dolphins off Bimini, The Bahamas, frequently execute a sharp turn before burying their rostrums in the sand. Based on data collected from 2012 to 2018, we report a significant right-side (left turn) bias in these dolphins. Out of 709 turns recorded from at least 27 different individuals, 99.44% ($n = 705$) were to the left (right side and right eye down) [$z = 3.275$, $p = 0.001$]. Only one individual turned right (left side and left eye down, 4/4 turns). We hypothesize that this right-side bias may be due in part to the possible laterization of echolocation production mechanisms, the dolphins' use of the right set of phonic lips to produce echolocation clicks, and a right eye (left hemisphere) advantage in visual discrimination and visuospatial processing.

## 1. Background

Lateralized behaviour is that in which an individual shows a significant and consistent bias in direction of movement, side orientation or use of a limb on one side of the body. Perhaps the most familiar example of lateralized behaviour is left- or right-handedness in humans, with approximately 90% of humans showing a right-side bias [1]. This preference already

exists *in utero*; approximately 9/10 fetuses suck their right thumb [1]. Lateralized behaviour at the population level, in which individuals within the population share a preference for the same side, is not limited to humans, and in fact exists in many vertebrate species [2]. For example, reindeer herds tend to circle in a counter-clockwise direction [3], and giraffes move their left leg first when beginning a splay stance [4]. Lateralization for limb preference can be found in a wide range of species, including lions, bats, toads, parrots and chickens [5]. Chimpanzees and gorillas show a significant right-hand bias, while orangutans show a significant left-hand bias [6].

Population-level lateralized behaviour suggests hemispheric specialization, in which different and specific functions are localized in one hemisphere of the brain [7]. In humans, language processing occurs primarily in the left cerebral hemisphere of the brain, while visuospatial processing occurs primarily in the right cerebral hemisphere [8,9]. This laterality in processing is linked to behavioural laterality; more than 95% of right-handed humans process language predominantly in the left hemisphere, while only approximately 70% of left-handed people process language in the left hemisphere [9]. Further evidence in support of hemispheric specialization lies in laterality in other cognitive functions. For example, humans and sheep share a left visual field bias (right hemisphere) for facial recognition [10]. Honeybees display a strong preference for using their right antenna rather than the left in social interactions [11]. Dogs wag their tail to the right when viewing positive stimuli (such as their owners) and wag to the left when viewing negative stimuli (such as a dominant, unfamiliar dog) [12].

Wild and captive cetacean species have also shown behavioural lateralization that suggests hemispheric specialization. Indo-Pacific bottlenose dolphins (*Tursiops aduncus*) show a bias in keeping conspecifics in their left visual field (right hemisphere) when initiating social pectoral fin rubbing [13], and beluga whale (*Delphinapterus leucas*) calves show the same predilection, keeping their mothers in their left visual field significantly more often than their right [14]. On the other hand, striped dolphins (*Stenella coeruleoalba*) show a right-eye preference for viewing unfamiliar objects [15] and common bottlenose dolphins (*T. truncatus*) perform significantly better in pattern discrimination, acquisition tasks and numerosity tasks when using the right eye (left hemisphere) [16–18].

Cetacean species (whales and dolphins) employ a wide variety of foraging techniques with many specializations appearing in different populations across the globe. Notably, many cetacean species demonstrate a significant right-side bias in foraging behaviours [7,19–21], a trend that is also seen in birds [22–25] and fish [26–28]. Population-level laterality with a right-side bias has been reported in bottom rolling in both grey whales (*Eschrichtius robustus*) [29] and humpback whales (*Megaptera novaeangliae*) with a ratio similar to the 9 : 1 right-hand bias found in humans [30]. Right-side bias in lunge feeding has been reported in orcas (*Orcinus orca*) [19], fin whales (*Balaenoptera physalus*), blue whales (*Balaenoptera musculus*), sei whales (*Balaenoptera borealis*) and Bryde's whales (*Balaenoptera brydei*) [20,31]. Dusky dolphins (*Lagenorhynchus obscures*) in populations in both New Zealand and Argentina circle clockwise around fish schools, keeping their right eyes and right sides towards their prey, when foraging [32]. Common bottlenose dolphins in South Carolina, the Northern Gulf of California and Georgia show a population-level bias during strand feeding, in which they intentionally beach themselves predominantly [33] or always [21,34] on their right side while cooperatively herding fish onto muddy river banks. Plume-feeding common bottlenose dolphins in the Florida Keys create a plume of mud and then turn and swim through the plume, usually with the right side of the body oriented into the water, to capture fish [35].

Common bottlenose dolphins off Bimini, The Bahamas and on White Sand Ridge, The Bahamas engage in a form of benthic foraging termed 'crater feeding' to obtain prey concealed in sandy substrate. Based on observations on White Sand Ridge [36] and observations off Bimini (JDK and KMS), prey off Bimini may include conger eel (family Congridae), wrasses (family Labridae) or clinids (family Clinidae). When crater feeding, a dolphin begins by swimming slowly approximately 1 m above the bottom, scanning and echolocating for prey buried in the sand using 'razor buzz' echolocation click trains. The dolphin then thrusts its rostrum into the sand to obtain its prey, leaving a crater behind [37]. We observed that prior to thrusting their rostrums into the sand, the dolphins in Bimini quickly stop their forward motion, often by making a rapid 90°–180° turn (figure 1). The dolphins usually turn to the left, keeping their right eyes and right sides closest to the bottom (see electronic supplementary material, video).

We hypothesize that, as in other foraging cetaceans, these dolphins have a right-side bias, and that this bias appears at the population level. In this study, we investigated whether there is consistent laterality in *Tursiops truncatus* turning behaviour during crater feeding both on an individual and population level.

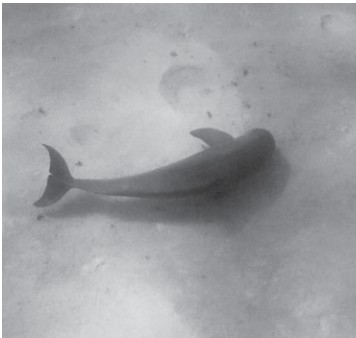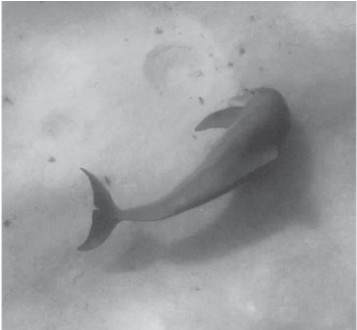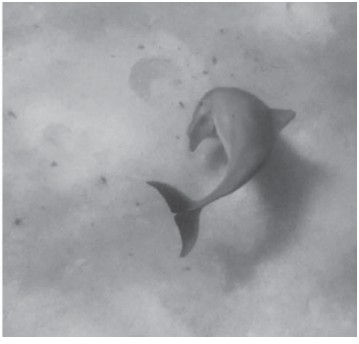

**Figure 1.** While crater feeding, a dolphin will intermittently execute a rapid 90°–180° turn, usually followed by thrusting its rostrum into the sand to obtain its prey.

## 2. Methods

### 2.1. Study area and study animals

Our study site was an area north and west of Bimini, The Bahamas, on the Great Bahama Bank. This site was used mainly during summer months to collect sighting, photographic, video and acoustic data of dolphin species including common bottlenose dolphins. This study area and species are part of a long-term longitudinal study dating back to 2001 (Dolphin Communication Project). At the time of this study, 129 individual bottlenose dolphins have been catalogued in Bimini based on distinguishing features [38]. Individual bottlenose dolphins can be distinguished based on notches, scarring and shape of the dorsal fin as well as scars throughout the body [39]. The bottlenose dolphins in this study are not provisioned, but are somewhat acclimated to boats and snorkelers through commercial swim-with-dolphin programmes, ecotourism expeditions, and long-term behavioural and population field studies. Most underwater observations of these dolphins are made during the benthic feeding behaviour known as crater feeding; this is when the dolphins are simultaneously not travelling and most tolerant of human presence. Additionally, these dolphins frequent shallow waters (less than 15 m) with excellent visibility ranging from 6 to 30+ m, depending on the weather and tide. These factors allowed the opportunity to record underwater observations of these study animals in close proximity.

### 2.2. Data collection and data analysis

The study site was accessed using local charter vessels (12.8 m Hatteras or 10.7 m Dakota). Surveys were typically conducted in the 4–6 h prior to sunset. A total of 337 boat surveys were conducted between 2012 and 2018 resulting in approximately 1580 effort hours. When dolphins were observed within vicinity of the moving vessel, vessel speed was reduced and the engine was put in neutral, allowing researchers and assistants to assess dolphin behaviour and, if appropriate, enter the water.

An encounter was defined as an in-water observation lasting over 1 min, in which the dolphin or dolphins were within visual range of the researcher. A new encounter was defined as at least an hour between sightings or when most dolphins were not present from the previous encounter.

Concurrent video and acoustic recordings were captured with several different underwater camera systems, including: (1) Canon HV30 in a custom underwater housing (The Sexton Company LLC, Salem, Oregon, USA) with SQ26 hydrophone input (Cetacean Research Technology, CRT, Seattle, Washington, USA) and a TASCAM DR-05 (Montebello, California, USA), (2) Canon Vixia HF20 in a Canon WP-V1 housing with custom hydrophones or SQ26-05 hydrophones from CRT, (3) Sony HDR-XR260 V in a custom housing with custom hydrophones [40] or (4) FujiFilm FinePix XP90. Research assistants took photographs that, along with surface images of dorsal fins, were later used for the identification of individuals.

During video analysis, a dolphin was considered to be crater feeding if it was swimming slowly, 1 m or less above the ocean floor, scanning the substrate. Scanning here refers to a dolphin swimming with its rostrum angled downward towards the substrate in the presence of audible razor buzzing. Although razor buzzes could not be localized to individual dolphins, these sounds could be heard throughout all foraging encounters. A 'turn' was defined as a dolphin rotating its body in a pinwheel motion

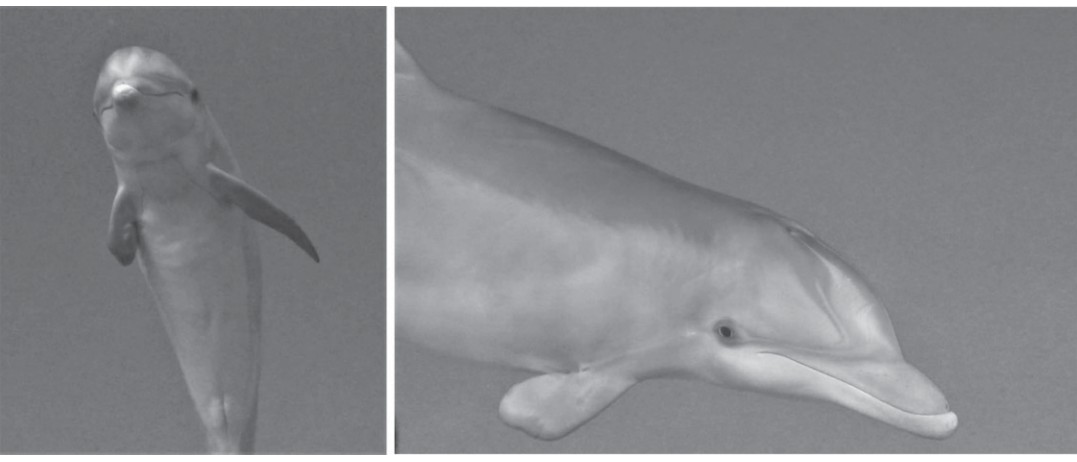

**Figure 2.** Tt134, which has an abnormally shaped right pectoral fin, was the only dolphin observed to turn right (left eye down) when crater feeding. All four observed turns from Tt134 were to the right.

during a crater feeding state (figure 1). The turn was noted as left or right. A left turn resulted in a dolphin's right eye facing the substrate; a right turn resulted in the left eye facing the substrate. Turn direction was scored by two independent observers viewing video footage. If the two observers disagreed on any observation, a third observer was employed to get agreement in two observers. When possible, dolphins were individually identified by scars and notches on dorsal and pectoral fins, and matched to an ID catalogue. Dolphins were often distanced far apart from each other during crater feeding, and thus not all dolphins present in an encounter were captured on video at the same time. Only turns captured on video were scored.

Binomial tests were run to test for significance in turn-direction preference at the individual level. This analysis included identified dolphins with at least 5 observed turns. With larger sample sizes, a binomial distribution approximates a normal distribution, and therefore $z$-scores were calculated for samples of greater than or equal to 10. Data were analysed using SPSS, Version 24. To determine whether there was a population-level right-side (left-turn) bias, a generalized linear mixed model (GLMM) with a binomial distribution was run using the glmer function from the package lme4 [41] in R [42]. Dolphin identities were included as a random effect within the model.

## 3. Results

A total of 26 encounters during which bottlenose dolphins were crater feeding were recorded across 2012–2018, totalling 649 min of recording. The average encounter length was 26 min (s.d. ± 17 min, range 3–65 min). The average crater feeding group size was 7.88 dolphins (s.d. ± 3.60 dolphins, median group size = 8 dolphins, range = 3–15 dolphins). We were able to identify, on average, approximately 26% of the dolphins in each encounter (s.d. ± 27%, range = 0%–83% of dolphins identified per encounter).

Seven hundred and nine turns were observed across encounters: 99.44% ($n = 705$) of all turns recorded were to the left (right eye down) and 0.56% ($n = 4$) were to the right. Dolphins within the population were significantly more likely to turn left than right with a predicted left-turn probability of 0.99 ($z = 3.28$, $p = 0.001$). Twenty-seven dolphins were individually identified from scars and notches on pectoral and dorsal fins, and always turned in the same direction; 26 always turned to the left (right side down) while only one dolphin, Tt134, always turned to the right (left side down; table 1). Tt134's right-turn bias was consistent across two encounters occurring in two different years. The four turns exhibited by Tt134 were the only right turns observed in all 709 observed turns. This individual has an abnormally shaped right pectoral fin (figure 2). The sample size of 27 identified individual dolphins that were recorded during crater feeding is likely to be an under-representation of the portion of the 129 catalogued dolphins that crater feed. Most of the dolphins observed during each crater feeding encounter were actively crater feeding, including both the 27 that could be individually recognized based on photo-ID and those that could not be identified. A subset of individually identified dolphins with 5 or more observations ($n = 23$) were tested for turn preference on an individual level; all 23 showed significant laterality, with a right-side (left-turn, right-eye down) bias (table 1).

**Table 1.** Laterality in turn behaviour in individual dolphins. Significant laterality in turn behaviour was found at the individual level, with all dolphins turning to the same side when turning during crater feeding. Note, Tt134 is the only individual observed to turn right, resulting in its left eye/side down.

| dolphin ID | total recording time | number of encounters and years | left turns/total turns | $p$-value | $z$ |
|---|---|---|---|---|---|
| Tt035 | 32:30 | 5: 2012, 2013, 2015, 2016 | 42/42 | <0.0001 | 6.33 |
| Tt040 | 24:27 | 5: 2014, 2015, 2016 | 39/39 | <0.0001 | 6.08 |
| Tt015 | 18:33 | 4: 2014, 2015, 2016, 2018 | 34/34 | <0.0001 | 5.66 |
| Tt031 | 16:26 | 1: 2015 | 30/30 | <0.0001 | 5.29 |
| Juvi A | 11:40 | 1: 2017 | 29/29 | <0.0001 | 5.20 |
| Calf A | 10:45 | 1: 2016 | 25/25 | <0.0001 | 4.80 |
| Tt039 | 19:15 | 4: 2013, 2014, 2015 | 21/21 | <0.0001 | 4.36 |
| Tt020 | 9:46 | 5: 2014, 2015, 2016, 2017 | 18/18 | <0.0001 | 4.01 |
| Tt407 | 8:44 | 1: 2018 | 17/17 | <0.0001 | 3.88 |
| Tt403 | 9:47 | 2: 2016 | 16/16 | <0.0001 | 3.75 |
| TtP | 4:46 | 1: 2017 | 14/14 | <0.0001 | 3.47 |
| Tt003 | 12:27 | 1: 2017 | 12/12 | <0.0001 | 3.18 |
| Tt038 | 5:41 | 2: 2016 | 12/12 | <0.001 | 3.18 |
| Tt080 | 3:38 | 1: 2017 | 12/12 | <0.001 | 3.18 |
| Tt094 | 7:58 | 3: 2017, 2018 | 8/8 | =0.004 | — |
| TtB | 3:31 | 1: 2015 | 7/7 | =0.008 | — |
| Tt109 | 5:16 | 1: 2018 | 7/7 | =0.008 | — |
| TtN | 2:07 | 1: 2016 | 7/7 | =0.008 | — |
| Tt1599 | 3:39 | 1: 2015 | 6/6 | =0.016 | — |
| TtL | 3:47 | 1: 2017 | 6/6 | =0.016 | — |
| Tt0 | 2:29 | 1: 2016 | 5/5 | =0.031 | — |
| Tt074 | 1:58 | 1: 2018 | 5/5 | =0.031 | — |
| Tt130 | 4:05 | 2: 2014, 2016 | 5/5 | =0.031 | — |
| Tt416 | 1:18 | 1: 2016 | 4/4 | — | — |
| Tt134 | 3:55 | 2: 2014, 2015 | 0/4 | — | — |
| TtQ | 0:12 | 1: 2015 | 1/1 | — | — |
| Tt028 | 1:23 | 1: 2015 | 1/1 | | |

## 4. Discussion

As hypothesized, common bottlenose dolphins in Bimini displayed lateralization in crater feeding both on an individual and at a population level with a strong right-side bias. Virtually all turns recorded both from the 27 individually identified dolphins and from unidentified dolphins (99.44% of turns) were to the left (right-side and right-eye down), and the analysis of 23 identified dolphins with 5 or more observations showed significant right-side (left-turn, right-eye down) bias. Only one dolphin turned to the right, and this right-turn behaviour was consistent in this dolphin across two separate encounters. It remains unclear if the malformed right pectoral fin of this individual or other factors may underlie this deviation from the common bias, particularly given that another individual (Tt407) also has a malformed right pectoral fin, and a third individual (Tt020) has no right pectoral fin, yet both Tt407 and Tt020 still exhibited right-side bias. Our findings are consistent with previous reports of a right-side bias in foraging behaviour in cetaceans.

Behavioural laterality in foraging behaviour has been documented in a wide variety of mammalian and non-mammalian species. Among cetaceans, examples include dusky dolphins, which keep their right eyes and right sides towards their prey when circling fish schools [32]. Common bottlenose

dolphins strand-feed and plume feed right-side down [21,33–35]. Grey whales and humpback whales show a population-level right-side bias when bottom rolling [29,30], and orcas, fin whales, blue whales, sei whales and Bryde's whales show a significant right-side bias when lunge feeding [19,20,31]. A right-side bias in feeding behaviours and foraging success has also been demonstrated in several species of birds [22–25] and fish [26–28].

A right-sided feeding bias in dolphins may have a physiological drive, where form meets function. Odontocetes demonstrate cranial asymmetry, which includes a leftward shift of the dorsal bones with larger bony nasofacial structures on the right side [43,44]. One explanation for odontocete cranial asymmetry proposes that skull asymmetry is a by-product of selection pressure for an asymmetrically positioned larynx; the larynx has been shifted to the left to provide room for a larger right pharyngeal food channel [45]. This larger right-sided food channel allows cetaceans to swallow larger prey items while limiting risk of these prey items becoming lodged around the intertwined larynx [45]. Another explanation for cranial asymmetry poses that this lateralization is linked to asymmetries in the soft tissues of the nasal complex involved in the generation and emission of echolocation clicks [43,44]. Dolphins have two structural complexes used to generate sound—the monkey lips dorsal bursae (MLDB)—one on each side of the head. The right MLDB is larger; the right set of phonic lips in particular are nearly twice as long as the left set [46].

The function of echolocation in particular may drive the right-side bias in odontocete foraging behaviours. Madsen *et al.* demonstrated that bottlenose dolphins produce echolocation clicks with the right set of phonic lips and whistles with the left [47]. When measuring the acoustic field on the forehead of echolocating common bottlenose dolphins, Au *et al.* [48] found slight asymmetry in signals, with higher amplitudes on the right side of the forehead. Perhaps the strongest example of a right-side swim bias in echolocating dolphins can be found in the Ganges or blind river dolphin (*Platanista gangetica*). Ganges River dolphins are functionally blind, with rudimentary eyes that most likely detect only light and dark [49,50]. These dolphins swim almost constantly on their sides [49,51,52], and continuously scan or echolocate as they swim [49,52]. In two reports of Ganges river dolphin behaviour, almost all dolphins observed swam continuously right-side down (12 out of 13 and 2 out of 3 dolphins observed). The remaining dolphins (1 out of 13 and 1 out of 3) swam continuously left-side down [49,52].

The left turn (right side and right eye down) bias observed in crater feeding common bottlenose dolphins may also be influenced by left hemisphere specialization in the processing of prey related sensory information, including visual and/or echolocation information. This left hemisphere specialization of processing of information related to prey has also been suggested in fish [19,53,54], birds [19,53,54] and other cetaceans [19,53,54]. Killer whales and humpback whales show a lateralization in feeding behaviours (specifically, lunging), but not in breaching, which suggests that the right-side bias in feeding behaviours specifically is driven by sensory lateralization [7,19]. In support of hemispheric differences in dolphins, widespread white matter laterality has been shown in the bottlenose dolphin brain, with most tracts exhibiting leftward structural asymmetries [55]. Bottlenose dolphins show significantly better performance in visuospatial tasks and numerosity tasks when using the right eye (left hemisphere) rather than the left eye (right hemisphere) [18,56]. Bottlenose dolphins also show superior performance when using the right eye in pattern discrimination tasks and in speed of acquisition when learning tasks [16,17]. The dolphin brain is capable of integrating visual and echoic information, enabling it to 'visualize' an object's shape through sound alone [57]. Thus, these crater feeding dolphins may be processing and integrating visual and echoic information in their left hemisphere.

Whether driven by anatomical structure or hemispheric specialization in sensory processing, a left-turn/right-side bias in crater feeding common bottlenose dolphins provides a strong demonstration of laterality in behaviour. Ringo has hypothesized that to minimize the delay in transmission time and facilitate information integration, increased hemispheric specialization may have evolved with increased brain size [58,59]. The bottlenose dolphin possesses the second largest brain-to-body mass ratio (EQ) of any mammalian species [60], thus we would expect a high degree of hemispheric specialization in the dolphin brain as well [59]. How this hemispheric specialization correlates with lateralized behaviour remains unclear. Advances in non-invasive brain imaging techniques hold the potential to further explore the link between behavioural laterality and hemispheric specialization and sensory and cognitive processing.

Ethics. Methods were observational only and adhered to local guidelines. This research was conducted under marine mammal research permits granted by The Bahamas Environment, Science and Technology Commission and the Department of Marine Resources, Ministry of Agriculture, Fisheries and Local Government, Nassau, The Bahamas.

Data accessibility. Data available from the Dryad Digital Repository: https://doi.org/10.5061/dryad.42jb0t4 [61].

Authors' contributions. J.D.K. contributed to conception and design, collected field data, analysed and interpreted data, and drafted the manuscript. S.Y.G. participated in data analysis and helped draft the manuscript. K.M.S. collected field data, contributed to conception and design, and critically revised the manuscript. D.L. critically revised the manuscript and contributed to conception and design. All authors gave final approval for publication and agree to be held accountable for the work performed therein.

Competing interests. We declare we have no competing interests.

Funding. This work was supported in part by Dolphin Communication Project (K.M.S.); Al Sweeting Jr (K.M.S.); Bimini Undersea (K.M.S.); The Animal Behaviour Society Cetacean Behaviour Award (J.D.K.); The CUNY Doctoral Student Research Grant (grant nos. 5, 7) (J.D.K.); The National Geographic Society Grant for Research and Exploration (D.R.) and the Hunter College Study Abroad Program (J.D.K. and D.R.).

Acknowledgements. Logistical support provided in part by Dolphin Communication Project (K.M.S.), Al Sweeting Jr and Bimini Undersea (K.M.S.). Statistical consultation provided by Laura Eierman.

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
