## [Reviewer comments · Royal Society Open Science]

Review History

RSOS-190929.R0 (Original submission)

Review form: Reviewer 1

Is the manuscript scientifically sound in its present form?

Yes

Are the interpretations and conclusions justified by the results?

Yes

Is the language acceptable?

Yes

Do you have any ethical concerns with this paper?

No

Have you any concerns about statistical analyses in this paper?

No

Recommendation?

Accept with minor revision (please list in comments)

Comments to the Author(s)

Here are suggestions for some grammatical errors in the text:

Page 3 Line 92- change it to "We hypothesize..."

Page 4 Line 95-I would suggest changing this to say "We investigated whether there is a consistent laterality..." because tested may suggest you did an experiment.

Page 5 Line 144 and Line 161-change to "...identified by scars and notches on dorsal and pectoral fins..."

Page 5 Line 155-157-Do you need this statement about other instances of crater feeding? It doesn't seem like they were counted, or if they were, explain more.

Page 5 Lines 158-165-You first say 709 turns but then reference 710 turns?

Page 5 Line 173-In the second sentence, are you reporting the results for the individual dolphins? This could be worded more clearly to relate to your hypothesis.

Review form: Reviewer 2**Is the manuscript scientifically sound in its present form?**

No

Are the interpretations and conclusions justified by the results?

No

Is the language acceptable?

Yes

Do you have any ethical concerns with this paper?

Yes

Have you any concerns about statistical analyses in this paper?

Yes

Recommendation?

Major revision is needed (please make suggestions in comments)

Comments to the Author(s)

Major comments:

The paper is an interesting addition to the literature on lateralization of behavior, especially in cetaceans. It adds to the growing body of literature. The introduction and discussion are extremely well-written and thorough. However, there are gaps that need to be filled on data reported and on statistics. The authors have independence and pseudo-replication issues with their data that need to be accounted for. By grouping all observations and all individuals together, the sample size is over-inflated. I think the results are valid, but possibly over-stated. Although I am not a statistician, it seems like some sort of repeated measures test should be done, as individuals were encountered multiple times. In addition, those multiple individuals were observed doing a behavior multiple times within one encounter. For example, Tt031 was encountered one time, but the data were treated as 30 different observations for the "population" level analysis.

On a slightly more minor note, the authors are inconsistent with taxonomic references to “bottlenose dolphins” throughout and should refer to the SMM taxonomy list (<https://www.marinemammalscience.org/species-information/list-marine-mammal-species-subspecies/>) by referring to *T. truncatus* as “common bottlenose dolphins” throughout to differentiate them from Indo-pacific bottlenose dolphins, which are also referenced in the paper.

Specific edits:

Abstract

l. 13: Common bottlenose dolphins (*Tursiops truncatus*)

Introduction

l. 59: common bottlenose dolphins (*T. truncatus*) – first reference to common bottlenose dolphins, yet don’t include species name.

ll. 73 and 77 – if add “common” then do not need species name after

l. 80 – add “Common” to differentiate from Indo-pacific

l. 92 – hypothesise (not hypothesis)

Methods

l. 103-104 – add common; remove *Tursiops truncatus*

l. 106 – this is confusing. So, did you use the 129 individuals identified? Will there be more? I think you can remove the “ongoing” part, as this is a finite study, I assume. Also, citation should be more broad/accessible for the audience, citing process of photo-ID, rather than an abstract. Finally, in background information, may be helpful to indicate if crater feeding is an individually specialized behavior – you have 129 individuals, but only saw 27 crater feeding. Is that because of a “small” sample size or are those the only individuals that crater feed?

l. 135 – add citation for photo-id for broader audience than just cetacean researchers

ll. 146-150 – need more detail. Sounds like did same simple test (binomial test) for all levels, yet have dependent (within encounters), pseudo-replicated (multiple encounters and observations/turns of same individuals) data.

On top of that mentioned above, seems like the data could get even more complicated. Methods and Results not clear on whether multiple individuals were seen within encounters. I assume there were (and by looking at the raw data supplement, it appears that there were multiple dolphins within encounters). So, is there lack of independence between individuals as well? (i.e., does one animal turning one way affect the animal next to it turning that way?)

Results

Would help to add more information; e.g., number of animals per encounter. Was it always just one? Were other animals in the area and not recorded? Were all animals that were crater feeding at once recorded? Would help understand if data are independent.

See comment above on Methods – not clear if 27 is a small sample size or if those are the only animals that crater feed.

Average number of turns per dolphin in each encounter (for those with multiple encounters)?

l. 156 – need semi-colon before however and a comma after

ll. 158-159 and l. 166 – report 709 and 710 turns. Need to correct. (and cross-check with addition in Table 1)

Discussion

Discussion is really well-written and thorough. Minor comments:

- I. 175 needs a comma before and
- II. 187 and 210 need to add "common" before bottlenose dolphin

Review form: Reviewer 3

Is the manuscript scientifically sound in its present form?

Yes

Are the interpretations and conclusions justified by the results?

Yes

Is the language acceptable?

Yes

Do you have any ethical concerns with this paper?

No

Have you any concerns about statistical analyses in this paper?

Yes

Recommendation?

Major revision is needed (please make suggestions in comments)

Comments to the Author(s)

This paper investigates the levels of lateralization during crater feeding, a foraging behaviour found in the bottlenose dolphins in The Bahamas. Based on over 700 observations from at least 27 individuals, the authors find a significant left-turn bias (right eye down) both on an individual- as well as population-level. They hypothesize that this lateralization could either be due lateralization in echolocation production, or a lateralized hemisphere specialization related to processing of sensory information.

This represents a neat, mostly well-written study in my eyes with interesting findings. There are, however, some weaknesses in the statistical analysis, as a binomial test is not appropriate to use with repeated measures of the same individuals. I nevertheless think this study warrants publication, and suggest it being published given (some of) my comments and concerns are addressed.

Detailed comments:

Abstract: well written, I endorse the cautious wording on the interpretation of the results

L11: To avoid repetition, I suggest starting the sentence with: 'At a population level, lateralization of behaviour....'

L20: I suggest adding the statistical results in brackets after '...down)'. It allows the reader to judge if the analysis is sound and the result can be trusted. For additional comments on statistical analyses, please see below as well.

Introduction: neatly written, giving various examples of lateralized behaviour in humans and non-human animals, before dealing with lateralization in cetaceans more specifically. I would suggest, however, to slightly change the structure of the last paragraphs, as it is basically already presenting results/discussion. I suggest the following changes:

L90+91: rephrase to: ‘... often by making a rapid 90- to 18- degree turn (Figure 1), keeping either their right or left eye and side close to the bottom (see supplemental video).’

L92-94: I suggest omitting both sentences, as it already presents the findings and discusses them.

L95+96: suggest to change to: ‘In the present study, we investigated whether there is consistent laterality in *Tursiops truncatus* turning behaviour during crater feeding both on an individual and population level.’

Methods:

106-111: I don’t understand how these two sentences connect (habituation and underwater observation of crater feeding). Also, it seems very long. I suggest breaking the sentence up in two parts, i.e., ‘The bottlenose dolphins in this study are not provisioned, but are somewhat acclimated to boats and snorkelers through commercial swim-with-dolphin programs, ecotourism expeditions, and long-term behavioural and population field studies.’

And have the ‘...with most underwater observations occurring during the benthic feeding behaviour known as crater feeding’ as a separate sentence. Could you also clarify the meaning of this sentence? I don’t think I follow what statement is being made with this.

119-120: suggest rephrasing ‘...resulting in approximately 1580 hours of looking for dolphins’ to ‘... resulting in approximately 1580 effort hours’.

136-138: suggest merging these sentences to: ‘During video analysis, a dolphin was considered to be crater feeding if it was swimming slowly, 1 metre or less above the ocean floor, scanning the substrate.

Also, does the ‘scanning’ refer to acoustic signals (echolocation) or was it inferred from head movements? Would be nice to clarify.

146-150: This is an issue that needs to be addressed: The use of the binomial test is appropriate to test for significance in turn-direction on an individual level, but not on the population level. A binomial test assumes independence of observations, which is violated if the same individuals are observed several times – this needs to be taken into account in the model.

I suggest using a generalized linear mixed effect model instead with a binomial error structure, where the dolphin IDs can be included as a random effect. I am not too familiar with SPSS and don’t know if such models can be run.

As an alternative, in R, I would suggest using the ‘lme4’ library with the ‘glmer’ function. In case you are not familiar with the R notations, this should work:

```
library(lme4)
```

```
model <- glmer(turn.direction~(1 | dolphin.ID), data=data.set, family="binomial")
```

```
summary(model)
```

The notation (1 | dolphin.ID) accounts for repeated measures of the same dolphins.

I strongly recommend to redo the population-level analysis and rewrite this paragraph accordingly, plus add the relevant information to the results and abstract.

156: comma after ‘however’

168: I suggest changing to ‘A subset of individually identified dolphins with 5 or more observations (n=23) were tested for turn preference on an individual level.’

Discussion:

172-174: To match changes suggested in the Intro, I suggest the following changes: 'Bottlenose dolphins in Bimini displayed lateralization during crater feeding both on an individual- as well as population level with a strong right-side bias.'

The rest of the Discussion reads very well, I could follow the argumentation, and the data supports the conclusions.

Decision letter (RSOS-190929.R0)

08-Aug-2019

Dear Dr Kaplan,

The editors assigned to your paper ("Behavioural Laterality in Foraging Bottlenose Dolphins (*Tursiops truncatus*)") have now received comments from reviewers. We would like you to revise your paper in accordance with the referee and Associate Editor suggestions which can be found below (not including confidential reports to the Editor). Please note this decision does not guarantee eventual acceptance.

Please submit a copy of your revised paper before 31-Aug-2019. Please note that the revision deadline will expire at 00.00am on this date. If we do not hear from you within this time then it will be assumed that the paper has been withdrawn. In exceptional circumstances, extensions may be possible if agreed with the Editorial Office in advance. We do not allow multiple rounds of revision so we urge you to make every effort to fully address all of the comments at this stage. If deemed necessary by the Editors, your manuscript will be sent back to one or more of the original reviewers for assessment. If the original reviewers are not available, we may invite new reviewers.

- Data accessibility

If you wish to submit your supporting data or code to Dryad (<http://datadryad.org/>), or modify your current submission to dryad, please use the following link:
<http://datadryad.org/submit?journalID=RSOS&manu=RSOS-190929>

- Competing interests

- Authors' contributions

- Acknowledgements

- Funding statement

Kind regards,
Lianne Parkhouse
Editorial Coordinator
Royal Society Open Science
openscience@royalsociety.org

on behalf of Dr Denise Greig (Associate Editor) and Kevin Padian (Subject Editor)
 openscience@royalsociety.org

Associate Editor's comments (Dr Denise Greig):

Three reviewers and I all agree that this manuscript is well written and of interest to the marine mammal community and journal readership. There are, however, issues with the statistical analyses of the data and data presentation (in the text and the results table). The reviewers offered several possible solutions, and I look forward to seeing a revised version of this manuscript.

Subject Editor's comments (Professor Kevin Padian):

Thanks for submitting. Please be sure to respond to all concerns of the reviewers with your resubmission. We look forward to the next version.

Reviewers' Comments to Author:

Reviewer: 1

Comments to the Author(s)

Here are suggestions for some grammatical errors in the text:

Page 3 Line 92- change it to "We hypothesize..."

Page 4 Line 95-I would suggest changing this to say "We investigated whether there is a consistent laterality..." because tested may suggest you did an experiment.

Page 5 Line 144 and Line 161-change to "...identified by scars and notches on dorsal and pectoral fins..."

Page 5 Line 155-157-Do you need this statement about other instances of crater feeding? It doesn't seem like they were counted, or if they were, explain more.

Page 5 Lines 158-165-You first say 709 turns but then reference 710 turns?

Page 5 Line 173-In the second sentence, are you reporting the results for the individual dolphins? This could be worded more clearly to relate to your hypothesis.

Reviewer: 2

Comments to the Author(s)

Major comments:

The paper is an interesting addition to the literature on lateralization of behavior, especially in cetaceans. It adds to the growing body of literature. The introduction and discussion are extremely well-written and thorough. However, there are gaps that need to be filled on data reported and on statistics. The authors have independence and pseudo-replication issues with their data that need to be accounted for. By grouping all observations and all individuals together, the sample size is over-inflated. I think the results are valid, but possibly over-stated. Although I am not a statistician, it seems like some sort of repeated measures test should be done, as individuals were encountered multiple times. In addition, those multiple individuals were observed doing a behavior multiple times within one encounter. For example, Tt031 was encountered one time, but the data were treated as 30 different observations for the "population" level analysis.

On a slightly more minor note, the authors are inconsistent with taxonomic references to “bottlenose dolphins” throughout and should refer to the SMM taxonomy list (<https://www.marinemammalscience.org/species-information/list-marine-mammal-species-subspecies/>) by referring to *T. truncatus* as “common bottlenose dolphins” throughout to differentiate them from Indo-pacific bottlenose dolphins, which are also referenced in the paper.

Specific edits:

Abstract

I. 13: Common bottlenose dolphins (*Tursiops truncatus*)

Introduction

I. 59: common bottlenose dolphins (*T. truncatus*) – first reference to common bottlenose dolphins, yet don’t include species name.

II. 73 and 77 – if add “common” then do not need species name after

I. 80 – add “Common” to differentiate from Indo-pacific

I. 92 – hypothesise (not hypothesis)

Methods

I. 103-104 – add common; remove *Tursiops truncatus*

I. 106 – this is confusing. So, did you use the 129 individuals identified? Will there be more? I think you can remove the “ongoing” part, as this is a finite study, I assume. Also, citation should be more broad/accessible for the audience, citing process of photo-ID, rather than an abstract.

Finally, in background information, may be helpful to indicate if crater feeding is an individually specialized behavior – you have 129 individuals, but only saw 27 crater feeding. Is that because of a “small” sample size or are those the only individuals that crater feed?

I. 135 – add citation for photo-id for broader audience than just cetacean researchers

II. 146-150 – need more detail. Sounds like did same simple test (binomial test) for all levels, yet have dependent (within encounters), pseudo-replicated (multiple encounters and observations/turns of same individuals) data.

On top of that mentioned above, seems like the data could get even more complicated. Methods and Results not clear on whether multiple individuals were seen within encounters. I assume there were (and by looking at the raw data supplement, it appears that there were multiple dolphins within encounters). So, is there lack of independence between individuals as well? (i.e., does one animal turning one way affect the animal next to it turning that way?)

Results

Would help to add more information; e.g., number of animals per encounter. Was it always just one? Were other animals in the area and not recorded? Were all animals that were crater feeding at once recorded? Would help understand if data are independent.

See comment above on Methods – not clear if 27 is a small sample size or if those are the only animals that crater feed.

Average number of turns per dolphin in each encounter (for those with multiple encounters)?

I. 156 – need semi-colon before however and a comma after

II. 158-159 and I. 166 – report 709 and 710 turns. Need to correct. (and cross-check with addition in Table 1)

Discussion

Discussion is really well-written and thorough. Minor comments:

- i. 175 needs a comma before and
- ii. 187 and 210 need to add "common" before bottlenose dolphin

Reviewer: 3

Comments to the Author(s)

This paper investigates the levels of lateralization during crater feeding, a foraging behaviour found in the bottlenose dolphins in The Bahamas. Based on over 700 observations from at least 27 individuals, the authors find a significant left-turn bias (right eye down) both on an individual- as well as population-level. They hypothesize that this lateralization could either be due lateralization in echolocation production, or a lateralized hemisphere specialization related to processing of sensory information.

This represents a neat, mostly well-written study in my eyes with interesting findings. There are, however, some weaknesses in the statistical analysis, as a binomial test is not appropriate to use with repeated measures of the same individuals. I nevertheless think this study warrants publication, and suggest it being published given (some of) my comments and concerns are addressed.

Detailed comments:

Abstract: well written, I endorse the cautious wording on the interpretation of the results

L11: To avoid repetition, I suggest starting the sentence with: 'At a population level, lateralization of behaviour....'

L20: I suggest adding the statistical results in brackets after '...down)'. It allows the reader to judge if the analysis is sound and the result can be trusted. For additional comments on statistical analyses, please see below as well.

Introduction: neatly written, giving various examples of lateralized behaviour in humans and non-human animals, before dealing with lateralization in cetaceans more specifically. I would suggest, however, to slightly change the structure of the last paragraphs, as it is basically already presenting results/discussion. I suggest the following changes:

L90+91: rephrase to: '... often by making a rapid 90- to 18- degree turn (Figure 1), keeping either their right or left eye and side close to the bottom (see supplemental video).'

L92-94: I suggest omitting both sentences, as it already presents the findings and discusses them.

L95+96: suggest to change to: 'In the present study, we investigated whether there is consistent laterality in *Tursiops truncatus* turning behaviour during crater feeding both on an individual and population level.'

Methods:

106-111: I don't understand how these two sentences connect (habituation and underwater observation of crater feeding). Also, it seems very long. I suggest breaking the sentence up in two parts, i.e., 'The bottlenose dolphins in this study are not provisioned, but are somewhat acclimated to boats and snorkelers through commercial swim-with-dolphin programs, ecotourism expeditions, and long-term behavioural and population field studies.'

And have the ,...with most underwater observations occurring during the benthic feeding behaviour known as crater feeding' as a separate sentence. Could you also clarify the meaning of this sentence? I don't think I follow what statement is being made with this.

119-120: suggest rephrasing '...resulting in approximately 1580 hours of looking for dolphins' to '... resulting in approximately 1580 effort hours'.

136-138: suggest merging these sentences to: 'During video analysis, a dolphin was considered to be crater feeding if it was swimming slowly, 1 metre or less above the ocean floor, scanning the substrate.

Also, does the 'scanning' refer to acoustic signals (echolocation) or was it inferred from head movements? Would be nice to clarify.

146-150: This is an issue that needs to be addressed: The use of the binomial test is appropriate to test for significance in turn-direction on an individual level, but not on the population level. A binomial test assumes independence of observations, which is violated if the same individuals are observed several times – this needs to be taken into account in the model.

I suggest using a generalized linear mixed effect model instead with a binomial error structure, where the dolphin IDs can be included as a random effect. I am not too familiar with SPSS and don't know if such models can be run.

As an alternative, in R, I would suggest using the 'lme4' library with the 'glmer' function. In case you are not familiar with the R notations, this should work:

```
library(lme4)
model <- glmer(turn.direction~(1 | dolphin.ID), data=data.set, family="binomial")
summary(model)
```

The notation (1 | dolphin.ID) accounts for repeated measures of the same dolphins.

I strongly recommend to redo the population-level analysis and rewrite this paragraph accordingly, plus add the relevant information to the results and abstract.

156: comma after 'however'

168: I suggest changing to 'A subset of individually identified dolphins with 5 or more observations (n=23) were tested for turn preference on an individual level.'

Discussion:

172-174: To match changes suggested in the Intro, I suggest the following changes: 'Bottlenose dolphins in Bimini displayed lateralization during crater feeding both on an individual- as well as population level with a strong right-side bias.'

The rest of the Discussion reads very well, I could follow the argumentation, and the data supports the conclusions.

Author's Response to Decision Letter for (RSOS-190929.R0)

See Appendix A.

RSOS-190929.R1 (Revision)

Review form: Reviewer 1

Is the manuscript scientifically sound in its present form?

Yes

Are the interpretations and conclusions justified by the results?

Yes

Is the language acceptable?

Yes

Do you have any ethical concerns with this paper?

No

Have you any concerns about statistical analyses in this paper?

No

Recommendation?

Accept as is

Comments to the Author(s)

This revised manuscript is very well-written and will be an important addition to the literature on laterality in cetaceans and other species.

Review form: Reviewer 2

Is the manuscript scientifically sound in its present form?

Yes

Are the interpretations and conclusions justified by the results?

Yes

Is the language acceptable?

Yes

Do you have any ethical concerns with this paper?

No

Have you any concerns about statistical analyses in this paper?

No

Recommendation?

Accept as is

Comments to the Author(s)

The introduction and discussion were already really well-written. Now, the methods and results match, as they are much more clear and statistically sound. Good job!

Review form: Reviewer 3

Is the manuscript scientifically sound in its present form?

Yes

Are the interpretations and conclusions justified by the results?

Yes

Is the language acceptable?

Yes

Do you have any ethical concerns with this paper?

No

Have you any concerns about statistical analyses in this paper?

No

Recommendation?

Accept as is

Comments to the Author(s)

I thank the authors for thoroughly incorporating my suggested changes. My major concern - the statistical analysis on the population level - has now also been appropriately dealt with. I am happy to accept the manuscript as is.

Decision letter (RSOS-190929.R1)

05-Nov-2019

Dear Dr Kaplan,

I am pleased to inform you that your manuscript entitled "Behavioural Laterality in Foraging Bottlenose Dolphins (*Tursiops truncatus*)" is now accepted for publication in Royal Society Open Science.

Royal Society Open Science operates under a continuous publication model (<http://bit.ly/cpFAQ>). Your article will be published straight into the next open issue and this will be the final version of the paper. As such, it can be cited immediately by other researchers.

As the issue version of your paper will be the only version to be published I would advise you to check your proofs thoroughly as changes cannot be made once the paper is published.

on behalf of Dr Denise Greig (Associate Editor) and Kevin Padian (Subject Editor)
openscience@royalsociety.org

Associate Editor Comments to Author (Dr Denise Greig):

As

Looks great! Thank you for doing that additional analysis as requested by the reviewers. I would delete "[z=3.275, p=0.001]" from the abstract as it refers to your probability estimate of making a left hand turn rather than the percentage of turns to the left.

Reviewer comments to Author:

Reviewer: 1

Comments to the Author(s)

This revised manuscript is very well-written and will be an important addition to the literature on laterality in cetaceans and other species.

Reviewer: 3

Comments to the Author(s)

I thank the authors for thoroughly incorporating my suggested changes. My major concern - the statistical analysis on the population level - has now also been appropriately dealt with. I am happy to accept the manuscript as is.

Reviewer: 2

Comments to the Author(s)

The introduction and discussion were already really well-written. Now, the methods and results match, as they are much more clear and statistically sound. Good job!

Appendix A

August 29, 2019

Dear Lianne Parkhouse, Dr Denise Greig, and Kevin Padian

On behalf of my coauthors and myself, I am submitting this response to referees to address and respond to comments provided by the reviewers, and to detail all adjustments made. We greatly appreciate the comments and suggestions that the reviewers provided. We are submitting a revised manuscript that incorporates the suggested changes. Please find below our responses to the reviewers and explanations of the changes noted in bold below after each of the reviewers' comments. The line numbers provided match the line numbers in the tracked-changes version of our resubmitted manuscript.

Sincerely,
J. Daisy Kaplan, PhD

Reviewers' Comments to Author:

Reviewer: 1

Comments to the Author(s)

Here are suggestions for some grammatical errors in the text:

Page 3 Line 92- change it to "We hypothesize..."

- **Line 94. Change made.**

Page 4 Line 95-I would suggest changing this to say "We investigated whether there is a consistent laterality..." because tested may suggest you did an experiment.

- **Line 97. Change made.**

Page 5 Line 144 and Line 161-change to "...identified by scars and notches on dorsal and pectoral fins..."

- **Line 155 and line 182. Change made.**

Page 5 Line 155-157-Do you need this statement about other instances of crater feeding? It doesn't seem like they were counted, or if they were, explain more.

- **We have deleted these lines. There were instances in which we noted possible crater feeding based on surface observations, but these data were not video recorded nor analyzed, and we thus agree with the reviewer that this statement about other instances of crater feeding is not needed here.**

Page 5 Lines 158-165-You first say 709 turns but then reference 710 turns?

- **Line 187. This was a typo, and should have read 709. We have changed this to 709. Thank you for catching this!**

Page 5 Line 173-In the second sentence, are you reporting the results for the individual dolphins? This could be worded more clearly to relate to your hypothesis.

- **Lines 198-200. The first two sentences have been combined and reworded for clarity, as follows: “As hypothesized, common bottlenose dolphins in Bimini displayed lateralization in foraging behaviour both on an individual and at a population level with a strong right-side bias.”**

Reviewer: 2

Comments to the Author(s)

Major comments:

The paper is an interesting addition to the literature on lateralization of behavior, especially in cetaceans. It adds to the growing body of literature. The introduction and discussion are extremely well-written and thorough. However, there are gaps that need to be filled on data reported and on statistics. The authors have independence and pseudo-replication issues with their data that need to be accounted for. By grouping all observations and all individuals together, the sample size is over-inflated. I think the results are valid, but possibly over-stated. Although I am not a statistician, it seems like some sort of repeated measures test should be done, as individuals were encountered multiple times. In addition, those multiple individuals were observed doing a behavior multiple times within one encounter. For example, Tt031 was encountered one time, but the data were treated as 30 different observations for the “population” level analysis.

- **We have redone our statistical analysis of lateralization at the population level using a generalized linear mixed effect model with dolphin IDs as random effects, as suggested by reviewer 3. This model avoids the issue of pseudo-replication.**

On a slightly more minor note, the authors are inconsistent with taxonomic references to “bottlenose dolphins” throughout and should refer to the SMM taxonomy list (<https://www.marinemammalscience.org/species-information/list-marine-mammal-species-subspecies/>) by referring to *T. truncatus* as “common bottlenose dolphins” throughout to differentiate them from Indo-pacific bottlenose dolphins, which are also referenced in the paper.

- **Thank you for this suggestion. We now refer to *T. truncatus* as common bottlenose throughout.**

Specific edits:

Abstract

l. 13: Common bottlenose dolphins (*Tursiops truncatus*)

- **Line 13. Changed to common bottlenose dolphins. We could not add the latin name, as we are at exactly 200 words with the additional requested stats added per reviewer 3.**

Introduction

l. 59: common bottlenose dolphins (*T. truncatus*) – first reference to common bottlenose dolphins, yet don’t include species name.

- **Line 60. We have now included the species name here.**

II. 73 and 77 – if add “common” then do not need species name after

- **Lines 75 and 79. Noted. ‘Common’ added and species name removed.**

I. 80 – add “Common” to differentiate from Indo-pacific

- **Line 82. Change made.**

I. 92 – hypothesise (not hypothesis)

- **Line 94. Change made.**

Methods

I. 103-104 – add common; remove *Tursiops truncatus*

- **Lines 106-107. Change made.**

I. 106 – this is confusing. So, did you use the 129 individuals identified? Will there be more? I think you can remove the “ongoing” part, as this is a finite study, I assume. Also, citation should be more broad/accessible for the audience, citing process of photo-ID, rather than an abstract. Finally, in background information, may be helpful to indicate if crater feeding is an individually specialized behavior – you have 129 individuals, but only saw 27 crater feeding. Is that because of a “small” sample size or are those the only individuals that crater feed?

- **Lines 109-112. Edits were made to clarify that the catalogue of 129 individuals was current at the time of this study and thus we feel the conference abstract is the most relevant citation. We also added a general/seminal reference for bottlenose dolphin photo-ID. We saw many dolphins crater feeding but we were only able to confirm the identity of 27 individuals based on photo-ID. To clarify this further, we have added average group size and average % of dolphins identified in each group (lines 173-176). We have also added the following sentences to further clarify this point (lines 188-192): “The sample size of 27 identified individual dolphins that were recorded during crater feeding is very likely to be an under-representation of the portion of the 129 catalogued dolphins that crater feed. Most of the dolphins observed during each crater feeding encounter were actively crater feeding, including both the 27 that could be individually recognized based on photo-ID and those that could not be identified.”**

I. 135 – add citation for photo-id for broader audience than just cetacean researchers

- **Line 112. This suggestion was integrated into the previous suggestion.**

II. 146-150 – need more detail. Sounds like did same simple test (binomial test) for all levels, yet have dependent (within encounters), pseudo-replicated (multiple encounters and observations/turns of same individuals) data.

- **Lines 169 -166. To address this issue, we have analyzed turn preference at the population level using a generalized linear mixed model (GLMM) with a binomial distribution, per reviewer 3’s recommendation. We have updated this paragraph in the methods section to reflect this change.**

On top of that mentioned above, seems like the data could get even more complicated. Methods

and Results not clear on whether multiple individuals were seen within encounters. I assume there were (and by looking at the raw data supplement, it appears that there were multiple dolphins within encounters). So, is there lack of independence between individuals as well? (i.e., does one animal turning one way affect the animal next to it turning that way?)

- **We saw many dolphins crater feeding, but we were only able to confirm 27 individuals based on photo-ID. To clarify this, we have added average group size and the average % of dolphins identified within each group (Lines 174-177). The dolphins did not forage or turn in unison nor were they in a proximity close enough to physically influence the directionality of another dolphin's turns – each dolphin continued on its own path while foraging and the dolphins were generally spaced out at least 1 metre and often 10+ metres. Nothing in their behavior suggested that any dolphin's orientation was influenced by another dolphin's behavior while foraging.**

Results

Would help to add more information; e.g., number of animals per encounter. Was it always just one? Were other animals in the area and not recorded? Were all animals that were crater feeding at once recorded? Would help understand if data are independent.

- **Several animals were present in each encounter, but not all animals were always recorded given the camera's field of view. To help clarify, we have now included average and median group size as well as the average % of dolphins we were able to ID in each encounter (lines 173-176). We have also added the following to the method section to further clarify (lines 156-158): “Dolphins were often distanced far apart from each other during crater feeding, and thus not all dolphins present in an encounter were captured on video at the same time. Only turns captured on video were scored.”**

See comment above on Methods – not clear if 27 is a small sample size or if those are the only animals that crater feed.

- **Please see our response above in the methods section.**

Average number of turns per dolphin in each encounter (for those with multiple encounters)?

- **We cannot give the average number of turns per encounter, as we conducted a series of focal follows during each encounter. During a focal follow, we followed one dolphin for several minutes, recording all behavior from this dolphin. We then switched focal follows to the next random dolphin in view to enable us to capture behavior of several dolphins in each encounter. When following one dolphin, most or all of the other dolphins present in the encounter were out of view of the camera. We do not have footage of any single dolphin from start to end of an encounter.**

l. 156 – need semi-colon before however and a comma after

- **This line is now removed.**

ll. 158-159 and l. 166 – report 709 and 710 turns. Need to correct. (and cross-check with addition in Table 1)

- **Line 187. This was a typo, thank you for catching this. The correct number is 709. This has now been corrected. We did not cross-check 709 with the numbers in Table 1, as table one lists the number of turns for individually identified dolphins only.**

Discussion

Discussion is really well-written and thorough. Minor comments:

- **Thank you!**

I. 175 needs a comma before and

- **Line 203. Change made.**

II. 187 and 210 need to add “common” before bottlenose dolphin

- **Lines 198 and 215. Change made.**

Reviewer: 3

Comments to the Author(s)

This paper investigates the levels of lateralization during crater feeding, a foraging behaviour found in the bottlenose dolphins in The Bahamas. Based on over 700 observations from at least 27 individuals, the authors find a significant left-turn bias (right eye down) both on an individual- as well as population-level. They hypothesize that this lateralization could either be due lateralization in echolocation production, or a lateralized hemisphere specialization related to processing of sensory information.

This represents a neat, mostly well-written study in my eyes with interesting findings. There are, however, some weaknesses in the statistical analysis, as a binomial test is not appropriate to use with repeated measures of the same individuals. I nevertheless think this study warrants publication, and suggest it being published given (some of) my comments and concerns are addressed.

- **We have redone our statistical analysis of lateralization at the population level using a generalized linear mixed effect model as suggested by this reviewer, and thank the reviewer for recommending this statistical test. We have also incorporated additional recommendations from this reviewer, as detailed in our responses below.**

Detailed comments:

Abstract: well written, I endorse the cautious wording on the interpretation of the results

L11: To avoid repetition, I suggest starting the sentence with: ‘At a population level, lateralization of behaviour....’

- **Lines 10-11. Change made.**

L20: I suggest adding the statistical results in brackets after ‘...down)’. It allows the reader to judge if the analysis is sound and the result can be trusted. For additional comments on statistical analyses, please see below as well.

- **Line 20. Statistical results have been added in brackets at the end of this sentence.**

Introduction: neatly written, giving various examples of lateralized behaviour in humans and non-human animals, before dealing with lateralization in cetaceans more specifically. I would suggest, however, to slightly change the structure of the last paragraphs, as it is basically already presenting results/discussion. I suggest the following changes:

L90+91: rephrase to: ‘... often by making a rapid 90- to 18- degree turn (Figure 1), keeping either their right or left eye and side close to the bottom (see supplemental video).’

- **Lines 91-93. We feel that it is more accurate to keep the wording of this sentence as is. The observation of a dolphin making right turn (left eye down) is extremely rare, and not consistent with what we observed. Notably, we observed that the dolphins consistently turned to the left, with the right eye down, during crater feeding. Based on this observation, we were surprised to find the one anomaly of one dolphin turning to the right when reviewing the video footage.**

L92-94: I suggest omitting both sentences, as it already presents the findings and discusses them.

- **Lines 94-97. We have deleted the sentence “A right-side population-level bias suggests that a hemispheric specialization governs this foraging behavior.” However, we feel that the first sentence “We hypothesize that, as in other foraging cetaceans, these dolphins have a right-side bias, and that this bias appears at the population level” should remain as is. There is a consistent right-side bias in foraging behavior among cetaceans, and we hypothesized that this same right-side bias would be found in this population and in this foraging behavior. The hypothesis is specifically that there is a right-side biased lateralization of behavior.**

*L95+96: suggest to change to: ‘In the present study, we investigated whether there is consistent laterality in *Tursiops truncatus* turning behaviour during crater feeding both on an individual and population level.’*

- **Lines 97-99. Change made.**

Methods:

106-111: I don’t understand how these two sentences connect (habituation and underwater observation of crater feeding). Also, it seems very long. I suggest breaking the sentence up in two parts, i.e., ‘The bottlenose dolphins in this study are not provisioned, but are somewhat acclimated to boats and snorkelers through commercial swim-with-dolphin programs, ecotourism expeditions, and long-term behavioural and population field studies.’

And have the ‘...with most underwater observations occurring during the benthic feeding behaviour known as crater feeding’ as a separate sentence. Could you also clarify the meaning of this sentence? I don’t think I follow what statement is being made with this.

- **Lines 112-118. We agree with the reviewer’s suggestion and have split this sentence into two sentences. We also added text (“this is when the dolphins are simultaneously not traveling and most tolerant of human presence.”) to clarify why the majority of bottlenose observations occur during crater feeding.**

119-120: suggest rephrasing ‘...resulting in approximately 1580 hours of looking for dolphins’ to ‘... resulting in approximately 1580 effort hours’.

- **Lines 126-127. Change made.**

136-138: suggest merging these sentences to: *‘During video analysis, a dolphin was considered to be crater feeding if it was swimming slowly, 1 metre or less above the ocean floor, scanning the substrate.*

- **Lines 143-145. Change made.**

Also, does the ‘scanning’ refer to acoustic signals (echolocation) or was it inferred from head movements? Would be nice to clarify.

- **Lines 145-148. We have added the following sentences to clarify this behavioral description; “Scanning here refers to a dolphin swimming with its rostrum angled downward towards the substrate in the presence of audible razor buzzing. Although razor buzzes could not be localized to individual dolphins, these sounds could be heard throughout all foraging encounters”**

146-150: This is an issue that needs to be addressed: The use of the binomial test is appropriate to test for significance in turn-direction on an individual level, but not on the population level. A binomial test assumes independence of observations, which is violated if the same individuals are observed several times – this needs to be taken into account in the model.

I suggest using a generalized linear mixed effect model instead with a binomial error structure, where the dolphin IDs can be included as a random effect. I am not too familiar with SPSS and don’t know if such models can be run.

As an alternative, in R, I would suggest using the ‘lme4’ library with the ‘glmer’ function. In case you are not familiar with the R notations, this should work:

```
library(lme4)
model <- glmer(turn.direction~(1|dolphin.ID), data=data.set, family="binomial")
summary(model)
```

The notation (1|dolphin.ID) accounts for repeated measures of the same dolphins.

I strongly recommend to redo the population-level analysis and rewrite this paragraph accordingly, plus add the relevant information to the results and abstract.

- **Thank you for this suggestion to redo the population-level analysis of our data. We have redone our statistics to determine if there is a significant population-level bias using a generalized linear mixed effect model. Information has been added to the abstract (line 20), methods (lines 163-166), and results (lines 180 – 181) and we thank the reviewer for this suggestion.**

156: comma after ‘however’

- **This sentence has been deleted.**

168: I suggest changing to ‘A subset of individually identified dolphins with 5 or more observations (n=23) were tested for turn preference on an individual level.’

- **Lines 193-194. Change made.**

Discussion:

172-174: To match changes suggested in the Intro, I suggest the following changes: 'Bottlenose dolphins in Bimini displayed lateralization during crater feeding both on an individual- as well as population level with a strong right-side bias.'

- **Lines 198-200. We have reworded this sentence as follows: "As hypothesized, common bottlenose dolphins in Bimini displayed lateralization in crater feeding both on an individual and at a population level with a strong right-side bias." We have kept the term 'foraging behavior' rather than change the term to 'crater feeding' as suggested by the reviewer because the emphasis here is on right-side bias in *foraging* behavior – this right side bias in foraging is a trend seen in many cetaceans as well as birds and fish.**

The rest of the Discussion reads very well, I could follow the argumentation, and the data supports the conclusions.

- **Thank you for this feedback!**